# ATTENTION IS ADVANTAGE: HOW THE WEAKER DEFEATS THE STRONGER THROUGH COOPERATION

## ABSTRACT

The phenomenon of weaker groups overcoming stronger opponents through cooperation in nature has inspired our exploration of cooperative-competitive mechanisms in multi-agent systems. In this work, we investigate emergent coordination policies in asymmetrical confrontations, focusing on how weaker agents collectively counter stronger opponents. The challenge of modeling such intricate interplay with a multilayer perceptron led us to adopt a Transformer architecture, which excels at capturing the complex, dynamic relationships between agents. We develop a two-phase curriculum training, with an attention-based strategy that effectively addresses policy training challenges in high-dimensional state spaces, and construct a scalable arena task that validates their effectiveness. Motivated by the Transformer's cooperative advantage, we utilize integrated gradients to attribute the contribution of attentions for each action dimension—thereby bridging attentions and behaviors and revealing how attentional dynamics scaffold collective superiority. This research provides a paradigm and an analytic approach for distributed collaboration in mixed adversarial systems.

## 1 INTRODUCTION

In nature, wolf packs demonstrate remarkable cooperative strategies to hunt large prey—such as bison or moose—that surpass any single wolf. These collaborative adversarial behaviors mirror key challenges in multi-agent reinforcement learning (MARL), where agents must balance competition and cooperation under asymmetric capabilities. While prior work has made strides in homogeneous cooperation or zero-sum competition, mixed cooperative-competitive settings with heterogeneous agents remain underexplored—particularly when weaker agents must learn to collaborate to defeat a stronger opponent. Developing agents capable of complex whole-body coordination, where physically weaker teams convert disadvantage into victory, remains an open challenge in the field.

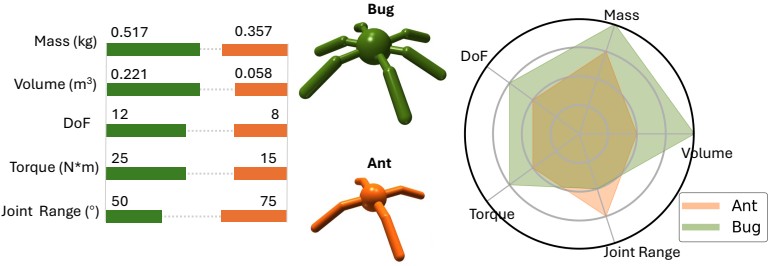

Figure 1: structural comparison of `ant` and `bug`. horizontal bar charts on the left illustrate five key metrics for each agent; the five-axis radar chart on the right visualizes these metrics after normalizing the raw values from the left, highlighting the `bug`'s greater bulk and torque and the `ant`'s agility.

A recent advance in attention-based MARL (Wen et al., 2022) offers promising tools for modeling agent interactions. Building on this advance, we enrich (i) *Embodiment* by representing capacity variance and morphological heterogeneity, and (ii) *Interpretability* by attributing attention flows among teammates and opponents. Specifically, we study a challenging physically embodied MARL

problem, where two heterogeneous teams of agents engage in adversarial confrontation. The task is inherently asymmetric: one team consists of more agents with smaller body size, lower mass, and weaker actuation capability, while the other team has fewer but stronger agents. This asymmetry creates a capability imbalance that requires sophisticated cooperation and body-based tactics. The environment includes two types of agents—`Ant` and `Bug` (see Figure 1). To illustrate the effectiveness of our method clearly and facilitate visualization for attribution analysis, we demonstrate three representative setups—2 `Ants` vs. 1 `Bug`, 3 `Ants` vs. 1 `Bug`, and 3 `Ants` vs. 2 `Bugs`—as shown in Figure 2, each of which can be easily generalized to the form of $m$ `Ants` vs. $n$ `Bugs`.

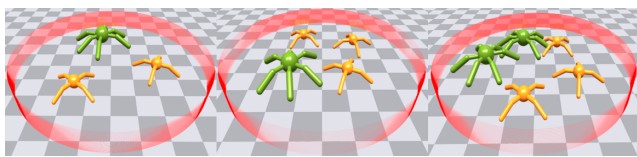

Figure 2: illustration of population settings in three scenarios: 2 `Ants` vs. 1 `Bug` (left), 3 `Ants` vs. 1 `Bug` (middle), and 3 `Ants` vs. 2 `Bugs` (right).

Based on this challenging asymmetric setting, we present three key innovations: (i) We propose an attention-aware curriculum learning framework that progressively trains agents from basic coordination to complex adversarial scenarios. (ii) We utilize gradient-based attribution methods for the embodied multi-agent systems (MAS), enabling qualitative analysis of how attention allocation correlates with emergent strategies. (iii) We design an embodied adversarial arena that captures the dynamics of asymmetric confrontation, where agents with varying physical capabilities must coordinate under competitive pressures, validating the proposed methods. Our approach not only bridges the embodiment gap in existing MARL methods but also provides the first systematic framework for analyzing attention-driven coordination in physically asymmetric scenarios. Online Page and Video: eama-sumo.github.io.

## 2 RELATED WORKS

**Learning-based Competition and Cooperation:** Artificial Intelligence (AI) powered policy has demonstrated significant capabilities in adversarial two-player tasks such as Go and Chess(Silver et al., 2016; 2017), and in complex multi-agent scenarios like Dota2(OpenAI et al., 2019), Star-Craft(Samvelyan et al., 2019; Yu et al., 2022), and simulated embodied competition tasks(Bansal et al., 2018; Huang et al., 2024a;b). Recently, embodied confrontational environments, like the Unitree World Robot Combat and competitive platform featuring quadruped robots MQE(Xiong et al., 2024) where agents physically interact within tasks like traversing bridges or wrestling, have further extended research into symmetric embodied adversarial games. Some works also study the asymmetric adversarial scenarios(Huang et al., 2024a). Furthermore, cooperation is another major aspect extensively researched in MAS. Some cooperative tasks are built around particle games, concentrating purely on synchronization and alignment in terms of position and velocity(Ma et al., 2022; Yang et al., 2018). Embodied cooperation by whole-body control is also introduced in recent years which brings new perspectives for swarm scalability and hierarchical task accomplishment(Hong et al., 2025). Meanwhile, higher-level cognition reasoning capabilities are investigated by applying large language models (LLMs) into teamwork(Liu et al., 2025; 2024; Zhang et al., 2024). However, there remains a notable gap in embodied tasks suitable for systematically studying heterogeneous asymmetric confrontation and collaboration in multi-agent contexts.

**Attention in MAS:** Attention-based sequence models are an effective substrate for multi-agent policy learning. Decision Transformer(Chen et al., 2021) demonstrates that causal self-attention can condition actions on returns-to-go and past trajectories, providing an off-policy alternative to value-based methods. Multi-Agent Transformer(Wen et al., 2022) adapts this idea to cooperative settings by factorising the joint value and letting cross-agent attention allocate credit among teammates, while MADT(Meng et al., 2021) shows that shared-parameter attention can support data-efficient transfer across tasks. Agent-Transformer Memory(Yang et al., 2022) further augments each agent with an external memory updated via recurrent attention, mitigating partial observability. Together, these studies underline how attention enables coordination, credit assignment, and knowledge reuse.

## 3 PROBLEM DEFINITION

Formally, the task is modeled as a two-team multi-agent Markov game. The first team contains $p$ agents and the second team contains $q$ agents. The game is defined by:

- A set of states $\mathcal{S}$ describing the environment and all agents' possible states.
- Action spaces $\mathcal{A}_\alpha = \{A_\alpha^1, A_\alpha^2, \ldots, A_\alpha^p\}$ and $\mathcal{A}_\beta = \{A_\beta^1, A_\beta^2, \ldots, A_\beta^q\}$ for the two teams $\alpha$ and $\beta$, where $A_\alpha^i$ and $A_\beta^j$ denote the action sets of the $i$-th and $j$-th agents in each team.
- A joint state transition function $\mathcal{T} : \mathcal{S} \times \mathcal{A}_\alpha \times \mathcal{A}_\beta \to \Delta(\mathcal{S})$, mapping the current state $\mathcal{S}$ and joint actions to a distribution over next states $\Delta(\mathcal{S})$.
- Reward functions $R : \mathcal{S} \times \mathcal{A}_\alpha \times \mathcal{A}_\beta \to r^i$ for each agent $i$, where $i \in \{1, \ldots, p + q\}$, dependent on the current state and joint actions.

To address this problem, equip each team with a fight policy $\pi_\alpha(\varphi)$ and $\pi_\beta(\phi)$, where $\varphi$ and $\phi$ denote the parameters of the team's policy. Each team learns its policy $\pi_\tau$, $\tau \in (\alpha, \beta)$ to generate effective cooperative and competitive strategies. At the beginning of each episode, agents act according to their respective team policies to maximize the expected cumulative team reward over a finite time horizon $T$. Specifically, the objective for team $\tau$ is to maximize the average return over all its agents:

$$J(\pi_\tau) = \frac{1}{|\mathcal{I}_\tau|} \sum_{i \in \mathcal{I}_\tau} \mathbb{E}_{\pi_\tau} \left[ \sum_{t=0}^{T} \gamma^t r_t^i \right],$$

where $\mathcal{I}_\tau$ denotes the set of agent indices in team $\tau$, $\gamma \in (0, 1]$ is the discount factor, and $r_t^i$ is the reward received by agent $i$ at time step $t$.

## 4 MIXED ASYMMETRIC COOPERATIVE-COMPETITIVE LEARNING

Embodied cooperative–competitive arenas oblige each agent to unify whole body control with strategic reasoning that balances intra-team collaboration and inter-team confrontation. To tackle this challenge, we introduce three complementary components: (1) a two-stage training curriculum that progressively guides agents from basic movement toward emergent confrontation strategies, and (2) application of the Multi-Agent Transformer architecture proposed by Wen et al. (2022), which leverages attention mechanisms to model both intra-team coordination and inter-team competition dynamics. (3) an attribution pipeline that maps transformer decisions back to semantically tagged observations, turning raw performance gains into interpretable insight.

### 4.1 TRAINING CURRICULUM

#### 4.1.1 CURRICULUM STAGE 1.

We use a simplified dense reward that encourages agents to move toward the center. Specifically, at each timestep, the agent receives a positive reward if its squared distance to the origin decreases compared to the previous timestep, and a negative reward otherwise. This dense reward guides agents to develop stable locomotion directed inward. Checkpoints with converged rewards and stable inward movement behaviors after 1000 epochs are selected to initialize Curriculum Stage 2.

#### 4.1.2 CURRICULUM STAGE 2.

In this stage, we use a single dense reward and discard the Curriculum Stage 1 reward. Training proceeds for 1000 epochs. Since agents can be eliminated during the game, at each timestep $t$, each agent $A_\alpha^i$ (or $A_\beta^j$) targets the first alive opponent in the opposing team. Let the opponent's position at timestep $t$ be $\mathbf{p}_t^{\beta,j}$ (or $\mathbf{p}_t^{\alpha,i}$), depending on the team. The agent's position at the previous timestep $t - 1$ is $\mathbf{p}_{t-1}^{\alpha,i}$ (or $\mathbf{p}_{t-1}^{\beta,j}$). We define the target vector $\mathbf{v}_t^{\text{target}} = \mathbf{p}_t^{\text{opponent}} - \mathbf{p}_{t-1}^{\text{agent}}$ and the movement-direction vector $\mathbf{v}_t^{\text{dir}} = \mathbf{p}_t^{\text{agent}} - \mathbf{p}_{t-1}^{\text{agent}}$, where $\mathbf{p}_t^{\text{agent}}$ and $\mathbf{p}_t^{\text{opponent}}$ denote the current positions of the agent and its chosen opponent. The reward at timestep $t$ is computed as $r_t = \max\left(0, \ \kappa \cdot \mathbf{v}_t^{\text{target}} \cdot \mathbf{v}_t^{\text{dir}}\right)$, with a scaling coefficient $\kappa > 0$ and the dot product "$\cdot$". Additionally, due

to the training in Curriculum Stage 1, agents seldom push opponents closer to the center, this being possible under this reward alone. This is because agents have learned to first occupy advantageous central positions before engaging opponents, making this reward aligned with the task dynamics.

## 4.2 Multi-Agent Transformer Architecture

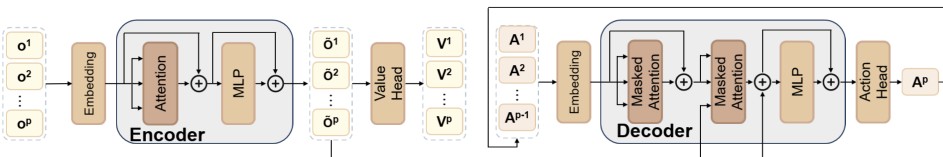

Figure 3: Overview of the Multi-Agent Transformer (MAT) architecture, which consists of a multi-agent observation encoder and an auto-regressive action decoder with masked attention.

We employ the Multi-Agent Transformer (MAT) to model interaction between agents. MAT follows an encoder-decoder design, where the encoder maps a sequence of agents' observations $(o^{i_1}, \ldots, o^{i_p})$ into latent representations $(\hat{o}^{i_1}, \ldots, \hat{o}^{i_p})$. Each encoder block consists of a self-attention module and an MLP, with residual connections to capture both individual features and inter-agent dependencies. The decoder takes the embedded joint action history $(a^{i_0}, \ldots, a^{i_{p-1}})$ and generates subsequent actions auto-regressively. Masked attention is used in the decoder to enforce causality, such that each agent's action $a^{i_p}$ only attends to previously generated actions $(a^{i_0}, \ldots, a^{i_{p-1}})$ and the encoded observations. An additional attention layer connects the decoder to the encoder outputs, conditioning action generation on latent observation representations.

To facilitate attribution, we use a simplified MAT configuration in this task: both the encoder and decoder consist of a single transformer block with one attention head. This setup has been found sufficient for modeling the cooperative-adversarial dynamics in our embodied setting while reducing computational overhead. The overall architecture is illustrated in Figure 3.

## 4.3 Interpreting Multi-Agent Transformer Decisions

### 4.3.1 Motivation for Further Analysis of Multi-Agent Transformer

While curriculum learning enhances agent performance, using a multi-agent transformer yields additional improvements over the multilayer perceptron (MLP), particularly in the complex 3Ants2Bugs scenario. Under MLP control, cooperation among the three Ants is sporadic and suboptimal, often involving ineffective formations, partial coordination, or isolated attacks. In contrast, transformer-controlled agents consistently form sophisticated, cooperative patterns such as stable triangular arrangements to restrict Bug mobility, coordinated stacking behaviors to impede Bug traversal, or linear group maneuvers that systematically compress Bugs' available space, increasing their elimination risk(Appendix A.1). The observed superiority suggests that the transformer's advantage arises from its structural capacity to model intricate agent interactions, which motivates a deeper investigation into the mechanisms underlying its enhanced performance.

### 4.3.2 Detailed Structure of Agent Observations

We are able to analyze at the dimension level rather than the vector level, since each input observation vector encodes interpretable information in four semantic categories: self states, teammate information, opponent information, and rule-specific indicators. Detailed dimensional meanings for the three population settings are given in Appendix A.7.

### 4.3.3 Tracing Action Decisions in Multi-Agent Transformer

To elucidate decision-making in our multi-agent transformer, we conduct a three-stage attribution analysis linking agents' actions back to observation dimensions (Figure 4). **Stage 1:** We apply integrated gradients to quantify how the encoder's self-attention pre-softmax weights influence the action. **Stage 2:** We attribute influential attention weights identified in Stage 1 back to specific di-

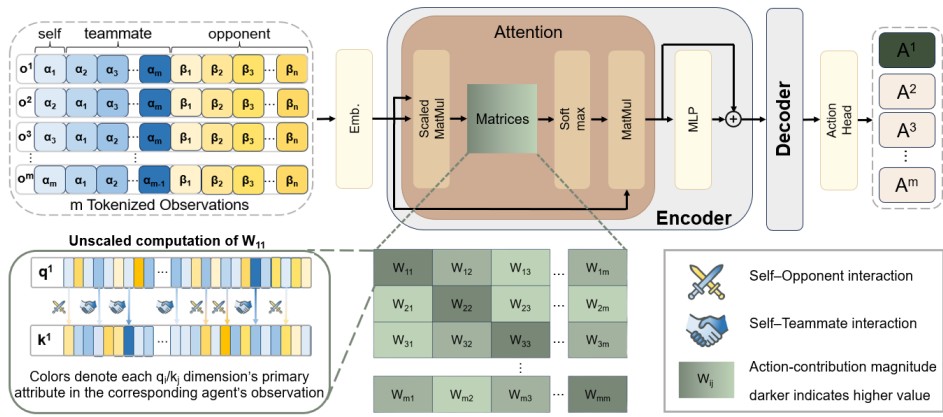

Figure 4: **Three-stage attribution pipeline.** Stage 1 attributes action-critical attention weights $w_{ij}$ via integrated gradients, darker cells in the lower-middle heatmap signal stronger influence. Stage 2 traces these weights to specific dimensions. Stage 3 attributes salient dimensions back to observation features. Color coding in the lower-left $q_i/k_j$ bars marks each dimension's primary attribute in the upper-left observation matrix, thereby revealing cooperative and adversarial information exchange.

mensions of the vectors. **Stage 3:** Finally, we attribute these salient dimensions back to the encoder's input observations, identifying which original observation features drive the agents' decisions.

**Stage 1: Attribution of pre-softmax attention weights to action dimensions**  We perform attribution on attention weights for each `Ant`'s action vector. We use a checkpoint trained on the complete two-stage curriculum and analyze a single decision step across 1024 parallel environments.

We apply the integrated gradients (IG) method to attribute the contribution of attention weights to each of the eight action dimensions. The baseline is set to the network initialization parameters (orthogonal initialization with gain 0.01), so as to avoid integration paths traversing unfamiliar data distributions that could yield unreliable outputs, and the integration path follows the standard linear interpolation from the baseline to the input. Here, the attention weights $w_{ij}$ are computed by the scaled dot-product between query and key embeddings in the encoder's self-attention mechanism, $w_{ij}^{(n)} = \frac{q_i^{(n)} \cdot k_j^{(n)}}{\sqrt{d_k}}$, where $d_k = 64$ is the hidden size, $n = 1, 2, \ldots, N$ is the index over parallel environments, and superscripts $(n)$ indicate quantities computed. For each `Ant`, to identify attention weights that consistently exhibit high contributions across a large number of parallel environments, the attribution score for an attention weight $w_{ij}$ is computed by:

$$\bar{\text{IG}}(w_{ij}) = \frac{1}{NS} \sum_{n=1}^{N} \sum_{s=1}^{S} \frac{\partial \hat{a}^{(n)}}{\partial w_{ij}^{(s,n)}} \cdot \Delta w_{ij}^{(s,n)},$$

where $\hat{a}$ denotes the action vector inferred by model, $S = 50$ is the number of interpolation steps.

**Stage 2: Dimensional contribution analysis of attention weights**  We investigate which specific dimensions of the 64-dimensional vectors contribute most to pre-softmax attention weights. This analysis focuses on attention weights $w_{ij}$ identified in Stage 1 as important. For each $w_{ij}$, we compute the average over $N$ environments, $\bar{w}_{ij} = \frac{1}{N} \sum_{n=1}^{N} w_{ij}^{(n)}$, for stability. Dimensions with larger values contribute more strongly to the corresponding attention weight, since $w_{ij}$ is calculated by first performing an element-wise product between $q_i$ and $k_j$ and then summing across all dimensions.

**Stage 3: Attribution from observations to query–key dimensions**  We attribute each observation dimension in $\text{obs}^{(n)}$ to every dimension of the attention weights $w_{ij}^{(n)}$. Building upon the high-contribution dimensions within these weights identified in Stage 2, we then focus on the semantic meanings of the observation features to which these dimensions are attributed. Consistent with Stage 1, we apply the Integrated Gradients method. For each $\text{obs}_i$, the baseline is set to the observation of `Ant` $i$ at the beginning of the episode, ensuring the baseline lies within a distribution familiar

to the model and that the attribution remains specific to the given agent in the given match. The integration path follows linear interpolation, while for quaternion dimensions we adopt spherical linear interpolation to respect their geometric structure. For each `Ant`, we average over $N$ environments and, by applying the product rule, obtain the attribution from observations as:

$$\overline{\text{IG}}(\text{obs}) = \frac{1}{NS\sqrt{d_k}} \sum_{n=1}^{N} \sum_{s=1}^{S} \left( k_j^{(s,n)} \cdot \frac{\partial \hat{q}_i^{(s,n)}}{\partial \text{obs}^{(s,n)}} \cdot \Delta \text{obs}^{(s,n)} + q_i^{(s,n)} \cdot \frac{\partial \hat{k}_j^{(s,n)}}{\partial \text{obs}^{(s,n)}} \cdot \Delta \text{obs}^{(s,n)} \right)$$

, where $\hat{q}_i^{(n)}$ and $\hat{k}_j^{(n)}$ denote the query and key vectors for the $i$-th query and $j$-th key in environment $n$, $S = 50$ is the number of interpolation steps with $s$ indexing each interpolation.

By leveraging the three-stage attribution pipeline, our interpretability framework reveals the Transformer's critical attention mechanisms in decision-making and shows how abstract attention scores map to concrete observation features; experimental results are presented in Section 5.4 and 5.5.

## 5 EXPERIMENTS AND RESULTS

### 5.1 ENVIRONMENT SETTINGS

We conduct experiments in a physically embodied sumo-style confrontation task built on IsaacGym, featuring cooperative–competitive interactions where two asymmetric teams compete while maintaining intra-team cooperation. The environment consists of a circular arena with an initial radius of 4.5 units, which gradually shrinks to a point over a time horizon $T = 1000$. An agent is eliminated once its distance from the origin exceeds the current arena radius. The arena hosts two types of agents—`Ant` and `Bug`—that differ in body morphology and control capabilities (see Figure 1).

To evaluate the effectiveness of our method under varying cooperative–adversarial dynamics, we consider three configurations with heterogeneous team sizes: 2 `Ants` vs. 1 `Bug`, 3 `Ants` vs. 1 `Bug`, and 3 `Ants` vs. 2 `Bugs` (Figure 2). These setups highlight asymmetric embodiment and strategic confrontation and can be easily generalized to $m$ `Ants` vs. $n$ `Bugs`. Experimental results for the 3`Ants` vs. 2`Bugs` configuration are presented in the main text, with the other two settings reported in the appendix. We use Proximal Policy Optimization (PPO) with Adam and a learning rate of 0.0001. The PPO clipping parameter is 0.2, the discount factor $\gamma = 0.99$, and the generalized advantage estimation parameter $\lambda = 0.95$. Training is parallelized across 1024 environments with 65,536 samples per batch and 4096 per minibatch.

### 5.2 EFFECT OF TRAINING CURRICULUM

We compare our curriculum-based method, which trains in two stages, with a non-curriculum baseline trained for 2000 epochs using a composite reward. The baseline combines sparse elimination-ranking rewards with dense terms that encourage central positioning, penalize joint-limit violations, suppress excessive movements, and discourage stillness via torque-based penalties. Both methods use simple multilayer perceptrons (MLPs) for action and value prediction.

To validate curriculum effectiveness, we test six settings with curriculum-trained and baseline (*non-curriculum*) `Ant` and `Bug` agents across three configurations: 2 `Ants` vs. 1 `Bug`, 3 `Ants` vs. 1 `Bug`, and 3 `Ants` vs. 2 `Bugs`. Experiments are grouped into *Curriculum Ablation*, where curriculum- and non-curriculum `Ants` face non-curriculum `Bugs`, and *Curriculum Robustness*, where they face curriculum-trained `Bugs`. Stage 1 rewards and Stage 2 win rates over epochs are shown in Figures 5 and 6. Curriculum-trained `Ants` consistently outperform baseline, showing stronger cooperation, confrontation, and longer survival. This advantage stems from the two-stage curriculum, which first stabilizes locomotion and then guides agents to interaction-centric skills aligned with task dynamics.

### 5.3 IMPACT OF POLICY NETWORK ARCHITECTURES UNDER CURRICULUM LEARNING

To further investigate policy network design under curriculum learning, we compare two representative architectures: a classical multilayer perceptron (MLP) and a multi-agent transformer. All agents are trained using the same curriculum described previously. We evaluate six settings by combining

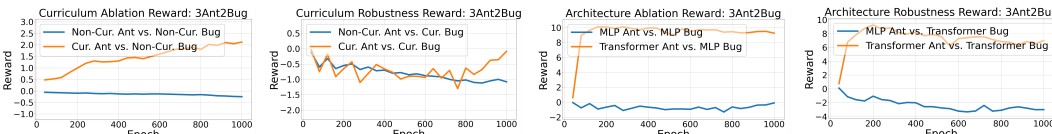

Figure 5: In the 3 `Ants` vs. 2 `Bugs` setting, across `Bug` training conditions, curriculum `Ants`' mean reward rises steadily, whereas non-curriculum `Ants` stay near zero or drop below 1.0. Transformer `Ants` reach rewards near 9 and remain positive, while MLP `Ants` stay below zero throughout.

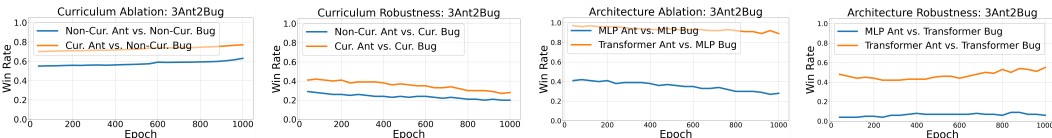

Figure 6: In the 3 `Ants` vs. 2 `Bugs` setting, across `Bug` training conditions, curriculum `Ants` outscore non-curriculum `Ants` by 0.2–0.3. Transformer `Ants` reach win rates of 0.9 vs. MLP `Bugs` and 0.55 vs. Transformer `Bugs`, while MLP `Ants` stall near 0.35 and 0.10.

network types for `Ant` and `Bug` agents across three configurations (2 `Ants` vs. 1 `Bug`, 3 `Ants` vs. 1 `Bug`, and 3 `Ants` vs. 2 `Bugs`). These settings fall into two categories: Architecture Ablation (MLP `Ant`/Transformer `Ant` *vs.* MLP `Bug`) and Architecture Robustness (MLP `Ant`/Transformer `Ant` *vs.* Transformer `Bug`). Figure 5 shows Stage 1 reward evolution, while Figure 6 depicts Stage 2 win-rate dynamics. Results consistently indicate that curriculum-trained transformer agents achieve higher performance and robustness than MLP counterparts, suggesting more expressive policies better capture the cooperative–adversarial dynamics of embodied multi-agent tasks.

## 5.4 THREE-STAGE ATTRIBUTION CASE STUDY

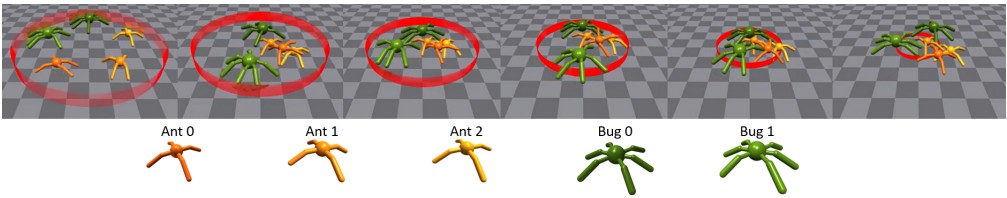

Figure 7: Snapshots uniformly sampled at regular intervals from a 3 `Ants` vs. 2 `Bugs` episode, illustrating how the `Ants` use a rod-shaped formation to block `Bug` cooperation and secure victory. Below, each agent (`Ant0/1/2`, `Bug0/1`) is shown with its corresponding color, providing a reference for tracking positions and roles across timesteps and for subsequent heatmap analyses.

Building on the methodological pipeline in Figure 4, this section illustrates a concrete case (Figure 7) linking `Ant` behavior to attention-attribution heatmaps(Appendix A.8). For six timesteps, three `Ants` are traced through the three-stage pipeline, while detailed heatmaps appear in the appendix. Section 5.5 then reveals strategies through heatmaps aggregated across large numbers of episodes.

We analyze the six snapshots sequentially. In Stage 1, attribution reveals that for all snapshots and all three `Ants`, $w_{ii}$ consistently receives the highest mean integrated-gradients attribution, exerting dominant influence over an agent's action. This does not imply that an `Ant` attends only to itself, since its observation already encodes information about teammates and opponents.

**For the first snapshot**, Stage 2 heatmaps highlight feature 35 for $w_{11}$, feature 55 for $w_{22}$, and features 19, 25, 46 for $w_{33}$. Stage 3 maps them respectively to $\text{obs}^1\{8, 9, 7, 112, 109\}$, $\text{obs}^2\{17, 7, 27, 109, 110\}$, and $\text{obs}^3\{25, 8, 27, 7, 111\}$. Appendix A.7 show these dimensions encode mainly self-velocity and `Bug` velocity. Since elimination is unlikely now, agents focus less on boundaries and instead emphasize speed to outpace `Bugs` and secure central ground.

**For the second and third snapshots** in the mid-game, Stage 2 heatmaps show features 58 and 42 dominate $w_{11}$, features 23 and 58 dominate $w_{22}$, and feature 29 dominates $w_{33}$. Stage 3 maps these to $\text{obs}^1$ dimensions $\{1, 94, 67, 40, 6\}$, $\text{obs}^2$ dimensions $\{93, 128, 5, 66, 40\}$, and $\text{obs}^3$ dimensions $\{5, 6, 0, 8, 119\}$. Appendix A.7 shows they correspond to: `Ant0` focusing on teammate/`Bug` relative positions plus posture, `Ant1` emphasizing both `Bugs` and posture, and `Ant2` stressing posture. These attributions highlight the stick-shaped formation through posture alignment and spacing, with `Ant1` leading near the center to split `Bugs`, while `Ant0` stabilizes posture as the central link.

**For the fourth snapshot**, Stage 2 heatmaps highlight feature 43 for $w_{11}$, feature 23 for $w_{22}$, and feature 25 for $w_{33}$. Stage 3 traces these to $\text{obs}^1 = 7, 93, 94, 128$, $\text{obs}^2 = 93, 128, 0, 66$, and $\text{obs}^3 = 0, 8, 122, 40$ (Appendix A.7). With the stick-shaped formation stabilized, posture adjustment is minimal. `Ant0` shifts to relative positioning against both `Bugs`, pressing harder as the shrinking boundary narrows their distance. `Ant1` sustains its central blocking role, while `Ant2`, farther from direct contact, attends mainly to positional cues of itself and teammates for alignment.

**For the fifth snapshot**, Stage 2 heatmaps highlight feature 42 for $w_{11}$, 23 for $w_{22}$, and 25 for $w_{33}$. Stage 3 attributes these to $\text{obs}^1\{7, 9, 29\}$, $\text{obs}^2\{93, 128, 66\}$, and $\text{obs}^3\{8, 29, 30\}$, with semantics given in Appendix A.7. `Ant0` and `Ant2` focus on boundary distances to prolong survival while exploiting velocity cues for last-moment inward pushes that shield `Ant1`. `Ant1`, by contrast, tracks relative positions with `Bugs`, sustaining pressure and preparing for the imminent 1-vs-2 clash.

**For the sixth snapshot**, with `Ant0` and `Ant2` eliminated, only `Ant1` remains. Stage 2 highlights feature 33 in $w_{22}$, and Stage 3 traces this to $\text{obs}_2$ dimensions $\{93, 0, 29, 154\}$. Appendix A.7 shows these correspond to `Ant1`'s boundary distance, its $x$-position, arena radius, and the `Bug`'s boundary distance. `Ant1` thus focuses on survival-critical cues—own and opponent boundary distances—while also attending to relative distance to the `Bug` and its position for central control.

## 5.5 Aggregated Three-Stage Attribution

This section aggregates integrated-gradients scores over all timesteps and averages across 1024 episodes to reveal the `Ants`' most common strategies in the `3Ants2Bugs` setting, visualized with heatmaps. Results for other team-size configurations appear in Appendix A.5.

For the Stage 1, we visualize the results with heatmaps (Figure 8); each cell represents the mean attribution from an attention weight to a specific action dimension. The main text reports three $8 \times 9$ heatmaps (8 action dimensions × 3 query × 3 key) for the `3Ants2Bugs` setting. In this population, Stage 1 attribution shows that the self-attention coefficient $w_{ii}$ commands the largest mean integrated-gradients score, dominating its own eight-dimensional action vector.



Figure 8: Stage 1 attribution heatmaps for the `3Ants2Bugs` task. Integrated-gradients map encoder self-attention weights (columns) to eight action dimensions (rows); in each `Ant`, $w_{ii}$ dominates.

For Stage 2, the heatmaps for the `3Ants2Bugs` task. Columns represent 64 feature dimensions, rows the attention weights $w_{ij}$. While features 26, 33, and 58 dominate $w_{11}$, features 11 and 26 are most salient for $w_{22}$, whereas features 26 and 58 dominate $w_{33}$.

For Stage 3, we observe three $64 \times 156$ heatmaps (Figure 10); rows index the 64 dimensions of $w_{ij}$ and columns the 156-dimensional observation vector. In the `3Ants2Bugs` case, features 26, 33, and 58 of $w_{11}$ show strongest attribution to the first `Ant`'s indices $\{66, 0, 40, 93, 7\}$ in $\text{obs}^1$, while features 11 and 26 of $w_{22}$ depend most on the second `Ant`'s indices $\{29, 154, 8, 67, 9\}$; features 26 and 58 of $w_{33}$ rely on indices $\{0, 40, 93, 67, 9\}$ for the third `Ant`. Appendix A.7 shows that these indices correspond to distinct semantic cues: for $\text{Ant}_0$, relative positions of both teammates and opponents as well as its own location; for $\text{Ant}_1$, distances to the border and to `Bugs` together

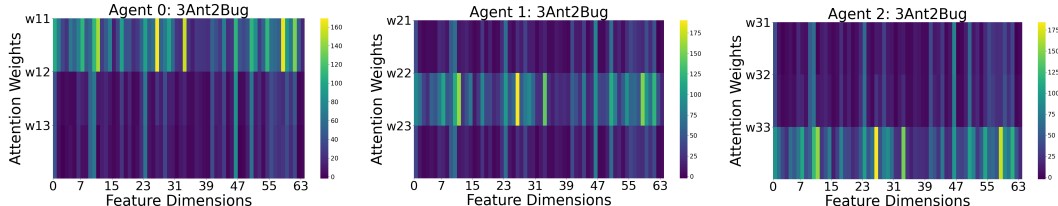

Figure 9: Stage 2 heatmaps for the `3Ants2Bugs` task. Columns denote 64 feature dimensions and rows the attention weights $w_{ij}$. Features 26, 33, and 58 dominate $w_{11}$; features 11 and 26 dominate $w_{22}$; and features 26 and 58 dominate $w_{33}$, respectively.

with velocity; and for $\text{Ant}_2$, relative positions to teammates. Overall, the three `Ants` consistently emphasize cooperative cues while monitoring their own border distances for survival and the `Bugs`' border distances to exploit elimination opportunities.

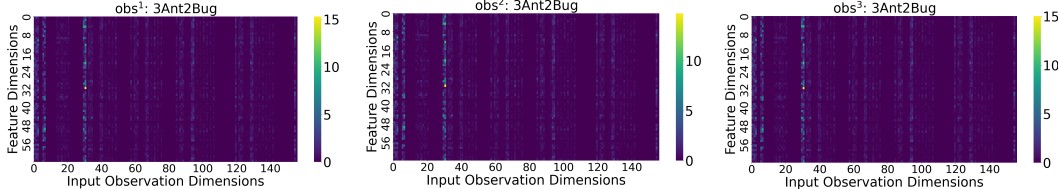

Figure 10: Stage 3 heatmaps for the `3Ants2Bugs` task. Columns index the 64 query–key features $c_{ij}$; rows enumerate the 156 input-observation dimensions. Feature 26 and 58 of $c_{11}$ mainly attribute to indices $\{29, 0, 30, 93, 86, 154, 39\}$ in $\text{obs}^1$, whereas feature 26 of $c_{22}$ and $c_{33}$ chiefly attribute to $\{29, 0, 30, 93, 86\}$ in $\text{obs}^2$ and $\text{obs}^3$, respectively.

### 5.6 Attribution Analysis: Cooperation, Competition, and Task Awareness

Due to each coordinate of the observation vector has an explicit physical or relational meaning—for instance, an `ant`'s absolute $(x, y)$ position or the relative offset to a teammate—we can read the three-stage attribution at the level of individual dimensions. It is noteworthy that, although Stage 1 assigns the largest attribution to the diagonal weight $w_{ii}$, this does not imply that agents exclusively rely on self-related information. This is because the input observation $\mathbf{o}_i$ already encodes information not only about agent $i$ itself but also about teammates and opponents. In fact, the attribution patterns fall into four distinct signal categories: *self* cues (agent's own $(x, y)$ coordinates and border distance); *co-operative* cues (relative $(x, y)$ offsets to teammates and teammates' border distances, facilitating formation control); *adversarial* cues (distances to each `Bug` and each `Bug`'s border distance, guiding pushing maneuvers or blocking escape routes); and *rule-specific* cues (agents' border distances, reflecting the Sumo elimination rule and influencing survival strategies). Thus, Attention supports intra-team coordination, inter-team competition, and task-aware positioning.

## 6 Conclusion

In this paper, an embodied multi-agent arena was designed to investigate the dynamics of asymmetric cooperative-competitive interactions, focusing particularly on how physically disadvantaged agents leverage coordination to overcome stronger opponents. Through a novel two-phase curriculum learning framework, agents developed intricate strategies—from stable locomotion to sophisticated collaborative adversarial maneuvers—demonstrating clear performance gains over traditional training methods. Moreover, by integrating gradient-based attribution techniques, the research offered interpretable insights into how the attention mechanisms underpinning the multi-agent transformer architecture facilitate tactical specialization and enhance agent coordination. Attention-driven models effectively capture complex intra-team and inter-team relationships, ultimately allowing weaker agents to evolve from individual actors into cohesive, strategically coordinated teams.

## ETHICS STATEMENT

This work does not involve human subjects, personally identifiable data, or sensitive content. It relies solely on simulated environments and publicly available benchmarks. We therefore believe that it does not raise any ethical concerns.

## REPRODUCIBILITY STATEMENT

We have taken extensive measures to ensure the reproducibility of our results. All implementation details, including the complete reward specification, curriculum design, and hyper-parameter settings, are given in Appendix A.2 and summarized in Table 1. Training dynamics and evaluation outcomes under different curricula and architectures are reported in Figures 12–15. The observation structure for each configuration is fully detailed in Appendix A.7, enabling exact reconstruction of input semantics. Case-study attribution heatmaps (Appendix A.8) provide transparency into the learned policies. Aggregated three-stage attribution heatmaps for the other population settings are compiled in Appendix A.5.

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

# A APPENDIX

## LLM USAGE STATEMENT

In preparing this manuscript, we made limited use of a large language model (LLM) to aid in polishing the writing. Specifically, the LLM was employed to improve clarity, conciseness, and grammatical accuracy of sentences, while all research ideas, experimental designs, analyses, and conclusions are entirely the work of the authors. No part of the technical content, data, or results was generated by the LLM. The authors take full responsibility for the final text.

## A.1 EMERGENT COOPERATIVE BEHAVIORS

The distinctive cooperation tactics learned by transformer-controlled Ants—ranging from coordinated stacking to encircling triangles or linear compressions—are vividly captured in Figure 11, which presents uniformly sampled frames from four separate episodes. Under an MLP policy, such maneuvers appear sporadic or incomplete, yet the transformer consistently choreographs each Ant's motion into high-order formations that severely constrain Bug mobility. The intriguing richness of these emergent patterns, as illustrated in Figure 11, naturally piqued our curiosity, prompting a closer investigation of the attention mechanisms that give rise to such sophisticated group behaviors.

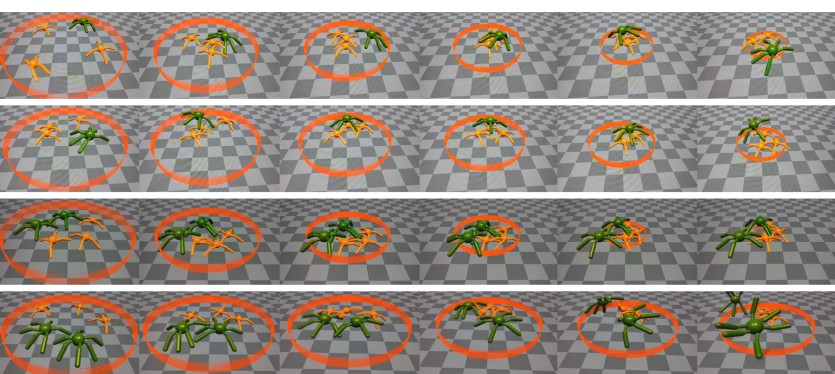

Figure 11: Snapshots uniformly sampled at regular intervals from individual episodes, arranged in four rows to illustrate four separate matches. The first row depicts Ants orchestrating coordinated stacking behaviors that initially impede Bug traversal, then coalesce into stable triangular formations that strip the Bug of footholds and induce a loss of balance. In the second row, group maneuvers systematically compress the Bug's available space, thereby elevating its risk of elimination. The third row reveals Ants forming encircling triangles to constrain the Bug's mobility, effectively trapping its legs and preventing escape. Finally, the bottom row captures three Ants synchronously exerting combined force to hurl two Bugs out of the arena.

## A.2 REWARD FUNCTION DESIGN

Each agent's per-step reward at timestep $t$ is decomposed as

$$r_{i,t} = r_{i,t}^{\text{dense}} + r_{i,t}^{\text{sparse}},$$

where $r_{i,t}^{\text{dense}}$ (the *Dense Shaping Reward*) encourages smooth, centered motion at each step, and $r_{i,t}^{\text{sparse}}$ (the *Sparse Outcome Reward*) provides a one-time ranking bonus upon termination.

DENSE SHAPING REWARD

At each timestep $t$, five dense components are aggregated:

$$r_{i,t}^{\text{dense}} = \underbrace{\alpha_{\text{center}} \exp\big(-\|\mathbf{p}_{i,t}\|_2\big)}_{\text{centering bonus}} + \underbrace{\alpha_{\text{limit}} \, n_{i,t}^{\text{limit}}}_{\text{joint-limit penalty}} + \underbrace{\alpha_{\text{act}} \, \|\boldsymbol{\tau}_{i,t}\|_2^2}_{\text{torque cost}}$$

$$+ \underbrace{\alpha_{\text{motion}} \exp\big(-\|\boldsymbol{\tau}_{i,t}\|_1\big)}_{\text{motion penalty}} + \underbrace{\alpha_{\text{flip}} \, \mathbf{1}\big(|\psi_{i,t}| < \tfrac{\pi}{4}\big)}_{\text{flip penalty}},$$

where $\mathbf{p}_{i,t} = (x_{i,t}, y_{i,t}) \in \mathbf{R}^2$ is agent $i$'s planar position at time $t$, $n_{i,t}^{\text{limit}}$ counts joints at their motion limits, $\boldsymbol{\tau}_{i,t} \in \mathbf{R}^m$ is the applied-torque vector, and $\psi_{i,t}$ is the yaw angle. The indicator is

$$\mathbf{1}\big(|\psi_{i,t}| < \tfrac{\pi}{4}\big) = \begin{cases} 1, & \text{if } |\psi_{i,t}| < \tfrac{\pi}{4}, \\ 0, & \text{otherwise.} \end{cases}$$

The hyperparameters are set to $\alpha_{\text{center}} = 20$, $\alpha_{\text{limit}} = -0.1$, $\alpha_{\text{act}} = -0.0025$, $\alpha_{\text{motion}} = -1.0$, $\alpha_{\text{flip}} = -10$.

SPARSE OUTCOME REWARD

When an episode terminates (at its final timestep $T$), each agent receives:

$$r_{i,t}^{\text{sparse}} = \text{reset}_{i,t} \times \Big( r_{\text{win}} - (k_{i,t} - 1) \, \tfrac{r_{\text{win}}}{agentNum} \Big),$$

where $\text{reset}_i \in \{0, 1\}$ indicates episode termination for agent $i$, $\text{reset}_{i,t} = 1$ only at termination for agent $i$, $k_{i,t}$ is its elimination rank at $t$, $r_{\text{win}} = 1000$, and agentNum the total agent count.

## A.3 CURRICULUM LEARNING REWARD

CURRICULUM STAGE I REWARD

At each timestep $t$, agent $i$ receives a centering reward that measures the reduction in distance to the arena center:

$$r_{i,t}^{\text{c1}} = \alpha_{\text{c1}}\big(\|\mathbf{p}_{t-1}^i\|_2 - \|\mathbf{p}_t^i\|_2\big),$$

where $\mathbf{p}_t^i = (x_t^i, y_t^i) \in \mathbf{R}^2$ denotes the agent's planar position, and $\alpha_{\text{c1}} > 0$ is the stage-specific coefficient (set to $\alpha_{\text{c1}} = 10$).

CURRICULUM STAGE II REWARD

In this stage, the Stage I reward is discarded and training proceeds for 1000 epochs. Each agent $A_i^\alpha$ (or $A_j^\beta$) targets the first alive opponent in the opposing team. Denote the agent's and opponent's positions by $\mathbf{p}_t^{\text{agent}}$, $\mathbf{p}_{t-1}^{\text{agent}}$, and $\mathbf{p}_t^{\text{opponent}}$. Define

$$\mathbf{v}_t^{\text{target}} = \mathbf{p}_t^{\text{opponent}} - \mathbf{p}_{t-1}^{\text{agent}}, \quad \mathbf{v}_t^{\text{dir}} = \mathbf{p}_t^{\text{agent}} - \mathbf{p}_{t-1}^{\text{agent}}.$$

The resulting reward is

$$r_{i,t}^{\text{c2}} = \max\big(0, \, \alpha_{\text{c2}} \, \mathbf{v}_t^{\text{target}} \cdot \mathbf{v}_t^{\text{dir}}\big),$$

with $\alpha_{\text{c2}} > 0$ the stage-specific coefficient (set to $\alpha_{\text{c2}} = 10$).

## A.4 HYPER-PARAMETER SETTINGS

In Table 1, we summarize the PPO hyper-parameters employed throughout our experiments. These include the learning rate, discount factor, GAE parameter, clipping threshold, and coefficients for entropy and critic losses, as well as settings for gradient clipping, minibatch configuration, and episode length. All values were chosen to maintain a stable learning process while ensuring sufficient policy exploration within the defined arena boundaries.

Table 1: Hyperparameters used for PPO training.

| Hyperparameter | Value |
| --- | --- |
| Learning rate | 0.0003 |
| Discount factor $\gamma$ | 0.99 |
| GAE $\lambda$ | 0.95 |
| PPO clip | 0.2 |
| Entropy coefficient | 0.0 |
| Critic coefficient | 2 |
| Max gradient norm | 1.0 |
| Horizon length | 64 |
| Minibatch size | $1024 \times$ agentNum |
| Mini-epochs | 4 |
| KL threshold | 0.008 |
| Episode length | 1000 |
| Number of environments | 1024 |
| Arena border space | 4.5 |

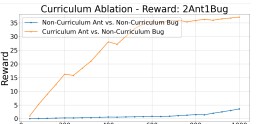 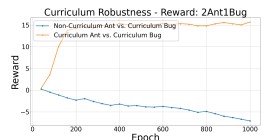 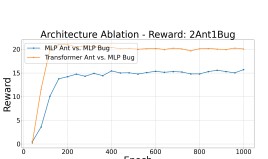 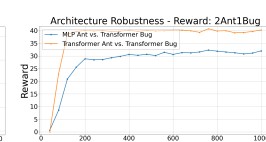

Figure 12: Learning curves in the 2 `Ants` vs. 1 `Bug` setting underscore the impact of curriculum design and network architecture: agents trained with a staged curriculum accrue over 30 reward points by epoch 1000—nearly 30 points more than non-curriculum peers—and preserve positive returns under shifts in `bug` behavior, whereas non-curriculum rewards slip into negative territory. Transformer-controlled `Ants` further accelerate progress, converging around 20 points in ablation tests (versus roughly 15 for MLPs) and driving rewards toward 40 in robustness trials, outpacing MLP counterparts stalled in the low-30s.

## A.5 EFFECT OF TRAINING CURRICULUM AND NETWORK ARCHITECTURE

Figure 12 shows the Stage 1 reward trajectories: the left panel contrasts a non-curriculum `ant` with a curriculum-trained `ant` as each faces the same `bug` (whether the `bug` itself was curriculum- or non-curriculum-trained), while the right panel compares two curriculum-trained architectures—MLP versus Transformer—each confronting its respective `bug` variant. Figure 13 presents the corresponding Stage 2 win-rate dynamics under the same match-ups. For the 3 `Ants` vs. 1 `Bug` configuration, the Stage 1 reward curves are shown in Figure 14, while the corresponding Stage 2 win rates appear in Figure 15. Curriculum-trained agents achieve both higher rewards and greater win rates than their non-curriculum counterparts; similarly, Transformer-based agents outperform MLP-based agents across these two metrics.

## A.6 ATTENTION ATTRIBUTION RESULTS

### A.6.1 2ANT1BUG

For the 2`Ants1Bug` configuration (Figures 16, 17, and 18), Stage 1 attribution indicates that the self-attention coefficient $w_{ii}$ achieves the highest mean integrated-gradients attribution across its eight-dimensional action vector, mirroring the pattern observed in the 3`Ants2Bugs` setting.

Stage 2 heatmaps (Fig. 17) arrange the 64 components of $w_{ij}$ as columns against the self-attention weights $w_{ij}$ as rows, revealing that the 46th dimension exerts the strongest influence on $w_{11}$ and $w_{22}$, respectively.

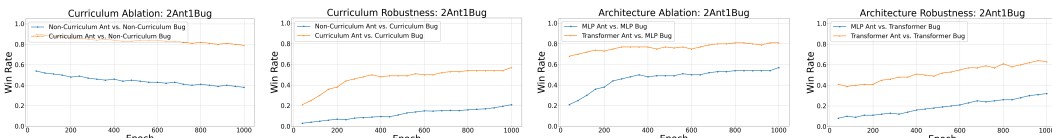

Figure 13: Learning curves across four conditions show that curriculum `Ants` maintain a steady 0.2–0.3 win-rate edge over non-curriculum peers—sustaining rates above 0.6 even when `Bug` behavior shifts (left panels). Transformer-powered `Ants` then climb toward 0.9 wins against MLP `Bugs` and about 0.55 versus Transformer `Bugs`, while MLP-based agents plateau near 0.35 and 0.10 (right panels).

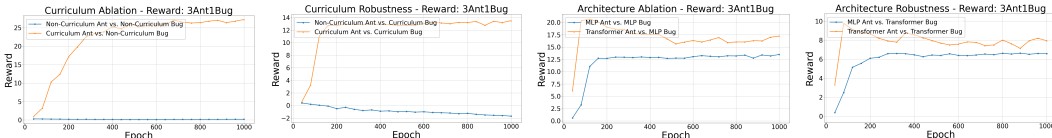

Figure 14: Reward trajectories in the 3 `Ants` vs 1 `Bug` setting reveal that without curriculum `Ants` barely progress, yet when trained through staged tasks they climb to about 25 points by epoch 1000 and retain roughly 13 points under shifted `Bug` behavior instead of sliding below zero. Transformer-driven `Ants` further outpace MLP controls—reaching near 18 points against MLP `Bugs` (versus 13 for MLPs) and holding around 9 points against Transformer `Bugs` (compared to 7 for MLP agents)

Stage 3 heatmaps (Fig. 18) plot the 64 dimensions $w_{ij}$ as columns against the 156-dimensional observation vector as rows, showing that the 46th feature of $w_{11}$ aligns with $obs^1$ indices $\{0,6,66,39,67\}$ (first ant's $x$-coordinate, orientation angle, $x$-offset to the `bug`, $x$-offset to the second `ant`, and $y$-offset to the `bug`) while the 46th feature of $w_{22}$ is most strongly associated with $obs^2$ indices $\{0,66,6\}$ (arena radius, first `ant`'s border distance, and $x$-offset to the `bug`).

### A.6.2 3ANT1BUG

For the 3`Ants1Bug` configuration (Figures 19, 20, and 21), we present the attribution analysis across all three stages. In Stage 1 (Fig. 19), the self-attention coefficient $w_{ii}$ yields the strongest average integrated-gradients attribution over its eight-dimensional action vector, mirroring the patterns seen in the 3`Ants2Bugs` and 2`Ants1Bug` tasks.

In Stage 2 (Fig. 20), columns enumerate the 64 components of $w_{ij}$ while rows correspond to the encoder self-attention weights $w_{ij}$; here, components 29, 44, and 60 of $\bar{c}_{11}$ dominate $w_{11}$, and the 17th component dominates $w_{22}$ and $w_{33}$.

Finally, in Stage 3 (Fig. 21), heatmaps plot the 64 dimensions $w_{ij}$ as columns against the 156-dimensional observation vector as rows: features 29, 44, and 60 of $w_{11}$ chiefly map to $obs^1$ indices $\{1, 0, 93, 29, 30, 119\}$ (first `ant`'s $y$- and $x$-coordinates, $x$-offset to the `bug`, distance to the border, arena radius, and `bug`'s border distance), whereas feature 17 of both $w_{22}$ and $w_{33}$ concentrates on indices $\{61, 30, 119\}$ in $obs^2$ and $obs^3$ (each `ant`'s $z$-coordinate, current border radius, and `bug`'s distance to the border).

Across the 2-`Ants` vs. 1-`Bug`, 3-`Ants` vs. 1-`Bug`, and 3-`Ants` vs. 2-`Bugs` settings, attention attribution consistently highlights four core signal categories—self-position and border distance, teammate offsets, adversary distances with their border metrics, and arena-boundary cues tied to elimination rules. By surfacing these physically and relationally meaningful features regardless of team composition, integrated-gradients attribution proves a practical tool for interpreting transformer-based multi-agent policies.

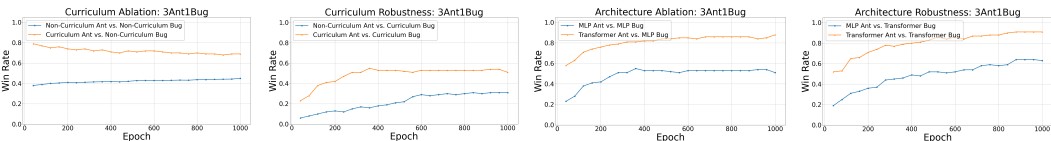

Figure 15: Win-rate curves in the 3 `Ants` vs. 1 `Bug` setting reveal that curriculum-trained `Ants` sustain roughly a 0.3 advantage—peaking near 0.75 versus 0.45 without curriculum and holding around 0.5 instead of 0.3 under `Bug` behavior shifts (left panels)—while Transformer-controlled `Ants` further lift performance, converging near 0.85 against MLP `Bugs` (versus $\tilde{0}.55$) and about 0.9 against Transformer `Bugs` (versus $\tilde{0}.65$) in architecture ablation and robustness tests (right panels).

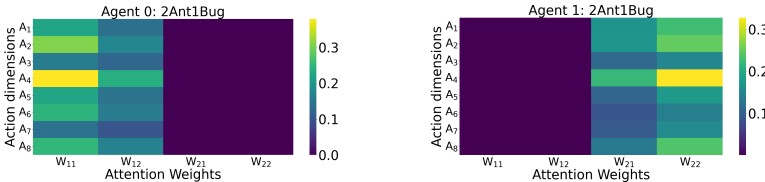

Figure 16: Stage 1 attribution heatmaps for the 2`Ants`1`Bug` task, in which integrated-gradients scores map encoder self-attention weights (columns) to the eight action dimensions (rows) and highlight the predominance of the diagonal weight $w_{ii}$ in each `Ant`.

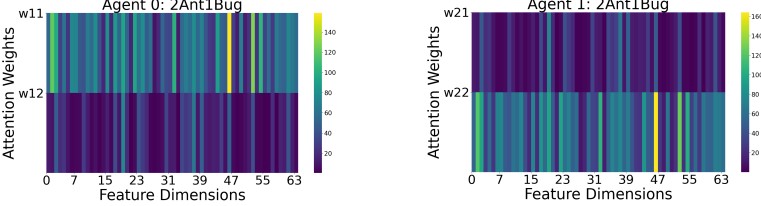

Figure 17: Stage 2 heatmaps for the 2`Ants`1`Bugs` task. Columns enumerate the 64 coordinates of $\bar{\mathbf{c}}_{ij}$, while rows correspond to the encoder self-attention weights $w_{ij}$. Feature 46 in $\bar{\mathbf{c}}_{11}$ contribute most to $w_{11}$; feature 46 of $\bar{\mathbf{c}}_{22}$ dominates $w_{22}$.

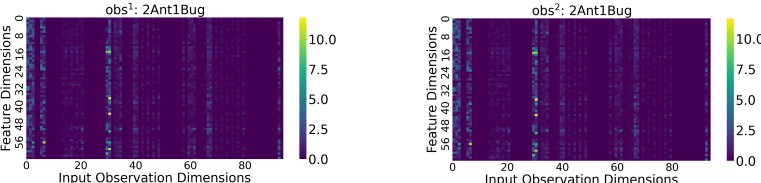

Figure 18: Stage 3 heatmaps for the 2`Ants`1`Bugs` task. Columns index the 64 query–key features $c_{ij}$; rows enumerate the 156 input-observation dimensions. Feature 46 of $c_{11}$ mainly attribute to indices $\{0, 6, 66, 39, 67\}$ in `obs`$^1$, whereas feature 46 of $c_{22}$ attribute to $\{0, 66, 6, 39, 1\}$ in `obs`$^2$.



Figure 19: Stage 1 attribution heatmaps for the 3`Ants`1`Bug` task: integrated-gradients scores link encoder self-attention weights (columns) to the eight action dimensions (rows), revealing the uniform predominance of the diagonal term $w_{ii}$ across all `Ant` agents.

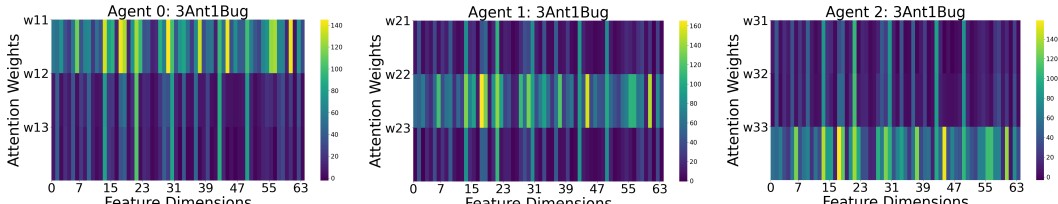

Figure 20: Stage 2 heatmaps for the 3Ants1Bugs task. Columns enumerate the 64 coordinates of $\bar{\mathbf{c}}_{ij}$, while rows correspond to the encoder self-attention weights $w_{ij}$. Feature 29, 44 and 60 in $\bar{\mathbf{c}}_{11}$ contribute most to $w_{11}$; feature 17 of $\bar{\mathbf{c}}_{22}$ dominates $w_{22}$; and feature 17 of $\bar{\mathbf{c}}_{33}$ dominates $w_{33}$.

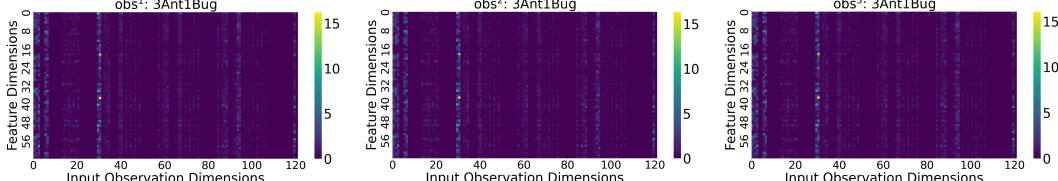

Figure 21: Stage 3 heatmaps for the 3Ants1Bugs task. Columns index the 64 query–key features $c_{ij}$; rows enumerate the 156 input-observation dimensions. Feature 29, 44 and 60 of $c_{11}$ mainly attribute to indices $\{1, 0, 93, 29, 30, 119\}$ in $\text{obs}^1$, whereas feature 17 of $c_{22}$ and $c_{33}$ chiefly attribute to $\{61, 30, 119\}$ in $\text{obs}^2$ and $\text{obs}^3$, respectively.

## A.7 OBSERVATION STRUCTURE DETAILS

Here we provide the explicit dimensional meanings of the observation vector for each experimental configuration (2 `Ants` vs. 1 `Bug`, 3 `Ants` vs. 1 `Bug`, and 3 `Ants` vs. 2 `Bugs`)

Table 2: 3`Ants` vs. 2`Bugs` observation dimensions and semantic meanings.

| INDEX RANGE | DESCRIPTION |
|---|---|
| 0–2 | Root position: `Self.x`, `Self.y`, `Self.z` |
| 3–6 | Root orientation quaternion: `Self.qw`, `Self.qx`, `Self.qy`, `Self.qz` |
| 7–9 | Linear velocity: `Self.vx`, `Self.vy`, `Self.vz` |
| 10–12 | Angular velocity: `Self.wx`, `Self.wy`, `Self.wz` |
| 13–20 | `Self` joint positions (8 DOF, scaled) |
| 21–28 | `Self` joint velocities (8 DOF, scaled) |
| 29 | Distance to border (`Self`) |
| 30 | Current border radius |
| 31 | Below termination height (`Self`) |
| 32–38 | `Ant1` root state (first 7 dims: pos3 + quat4) |
| 39–40 | Relative displacement: `Self.xy` − `Ant1.xy` |
| 41–48 | `Ant1` joint positions (8 DOF, scaled) |
| 49–56 | `Ant1` joint velocities (8 DOF, scaled) |
| 57 | `Ant1` distance to border |
| 58 | `Ant1` below termination height |
| 59–65 | `Ant2` root state (first 7 dims) |
| 66–67 | Relative displacement: `Self.xy` − `Ant2.xy` |
| 68–75 | `Ant2` joint positions (8 DOF, scaled) |
| 76–83 | `Ant2` joint velocities (8 DOF, scaled) |
| 84 | `Ant2` distance to border |
| 85 | `Ant2` below termination height |
| 86–92 | `Bug0` root state (first 7 dims) |
| 93–94 | Relative displacement: `Self.xy` − `Bug0.xy` |
| 95–106 | `Bug0` joint positions (12 DOF, scaled) |
| 107–118 | `Bug0` joint velocities (12 DOF, scaled) |
| 119 | `Bug0` distance to border |
| 120 | `Bug0` below termination height |
| 121–127 | `Bug1` root state (first 7 dims) |
| 128–129 | Relative displacement: `Self.xy` − `Bug1.xy` |
| 130–141 | `Bug1` joint positions (12 DOF, scaled) |
| 142–153 | `Bug1` joint velocities (12 DOF, scaled) |
| 154 | `Bug1` distance to border |
| 155 | `Bug1` below termination height |

Table 3: 3Ants vs. 1Bug observation dimensions and semantic meanings.

| INDEX RANGE | DESCRIPTION |
|---|---|
| 0–2 | Self root position: pos_x, pos_y, pos_z |
| 3–6 | Self root quaternion: quat_w, quat_x, quat_y, quat_z |
| 7–9 | Self linear velocity: lin_vel_x, lin_vel_y, lin_vel_z |
| 10–12 | Self angular velocity: ang_vel_x, ang_vel_y, ang_vel_z |
| 13–20 | Self 8 joint positions (scaled to [1, 1]) |
| 21–28 | Self 8 joint velocities (scaled by dofVelocityScale) |
| 29 | Distance to border |
| 30 | Current border radius |
| 31 | Below termination height (bool → float) |
| 32–38 | Ant1 root state (pos_x, pos_y, pos_z, quat_w, quat_x, quat_y, quat_z) |
| 39–40 | Relative displacement: self.xy − Ant1.xy |
| 41–48 | Ant1 joint positions (8 DOF, scaled) |
| 49–56 | Ant1 joint velocities (8 DOF, scaled) |
| 57 | Ant1 distance to border |
| 58 | Ant1 below termination height |
| 59–65 | Ant2 root state (pos_x, pos_y, pos_z, quat_w, quat_x, quat_y, quat_z) |
| 66–67 | Relative displacement: self.xy − Ant2.xy |
| 68–75 | Ant2 joint positions (8 DOF, scaled) |
| 76–83 | Ant2 joint velocities (8 DOF, scaled) |
| 84 | Ant2 distance to border |
| 85 | Ant2 below termination height |
| 86–92 | Bug0 root state (pos_x, pos_y, pos_z, quat_w, quat_x, quat_y, quat_z) |
| 93–94 | Relative displacement: self.xy − Bug0.xy |
| 95–106 | Bug0 joint positions (12 DOF, scaled) |
| 107–118 | Bug0 joint velocities (12 DOF, scaled) |
| 119 | Bug0 distance to border |
| 120 | Bug0 below termination height |

Table 4: 2Ants vs. 1Bug observation dimensions and semantic meanings.

| INDEX RANGE | DESCRIPTION |
|---|---|
| 0–2 | Self root position: pos_x, pos_y, pos_z |
| 3–6 | Self root quaternion: quat_w, quat_x, quat_y, quat_z |
| 7–9 | Self linear velocity: lin_vel_x, lin_vel_y, lin_vel_z |
| 10–12 | Self angular velocity: ang_vel_x, ang_vel_y, ang_vel_z |
| 13–20 | Self 8 joint positions (scaled to [1, 1]) |
| 21–28 | Self 8 joint velocities (scaled by dofVelocityScale) |
| 29 | Distance to border |
| 30 | Current border radius |
| 31 | Below termination height (bool → float) |
| 32–38 | Other Ant root state (pos_x, pos_y, pos_z, quat_w, quat_x, quat_y, quat_z) |
| 39–40 | Relative displacement: self.xy − ant.xy |
| 41–48 | Other Ant joint positions (8 DOF, scaled) |
| 49–56 | Other Ant joint velocities (8 DOF, scaled) |
| 57 | Other Ant distance to border |
| 58 | Other Ant below termination height |
| 59–65 | Bug0 root state (pos_x, pos_y, pos_z, quat_w, quat_x, quat_y, quat_z) |
| 66–67 | Relative displacement: self.xy − bug0.xy |
| 68–79 | Bug0 joint positions (12 DOF, scaled) |
| 80–91 | Bug0 joint velocities (12 DOF, scaled) |
| 92 | Bug0 distance to border |
| 93 | Bug0 below termination height |

## A.8 CASE STUDY HEATMAPS

Building on the methodological pipeline in Figure 4 and the illustrative case in Figure 7, this subsection provides the detailed Stage 1–3 attribution heatmaps for each of the three `Ant` agents. For six representative snapshots sampled across a single `3Ants` vs. `2Bugs` episode, we visualize how attention-to-action mappings evolve, complementing the sequential analysis in Section 5.4. These heatmaps ground the case study with fine-grained visualizations that reveal the specific observation dimensions driving policy decisions.

Figures 24–22 show Stage 1–3 heatmaps for `Ant0`, tracing its posture stabilization and central-link role in the stick-shaped formation. Figures 27–25 present corresponding maps for `Ant1`, highlighting its consistent central blocking and leadership role as the formation anchor. Finally, Figures 30–28 detail attributions for `Ant2`, which functions as the rear-supporting agent, attending more to alignment and boundary cues in later timesteps. Together, these nine figures provide transparency into the learned policies by linking agent roles to their evolving attention focus.

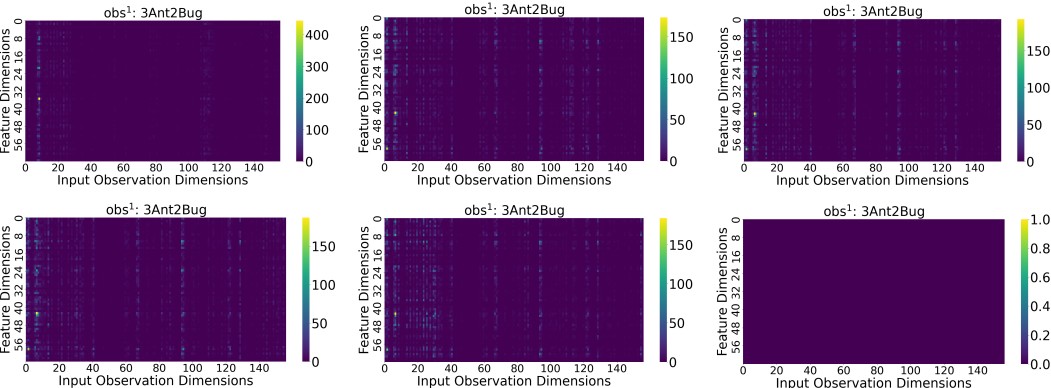

Figure 22: Stage 3 attribution heatmaps for `Ant0` in the `3Ants` vs. `2Bugs` case. The first row corresponds to snapshots 1–3, the second row to snapshots 4–6, tracing how action-level attention weights evolve across timesteps.

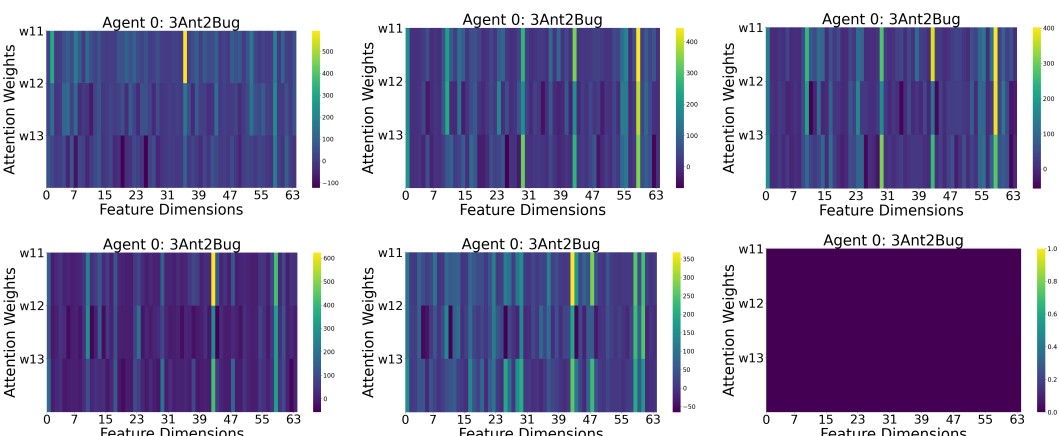

Figure 23: Stage 2 attribution heatmaps for `Ant0`. The first row shows snapshots 1–3 and the second row snapshots 4–6, revealing which feature dimensions contribute most to attention weights over the episode.

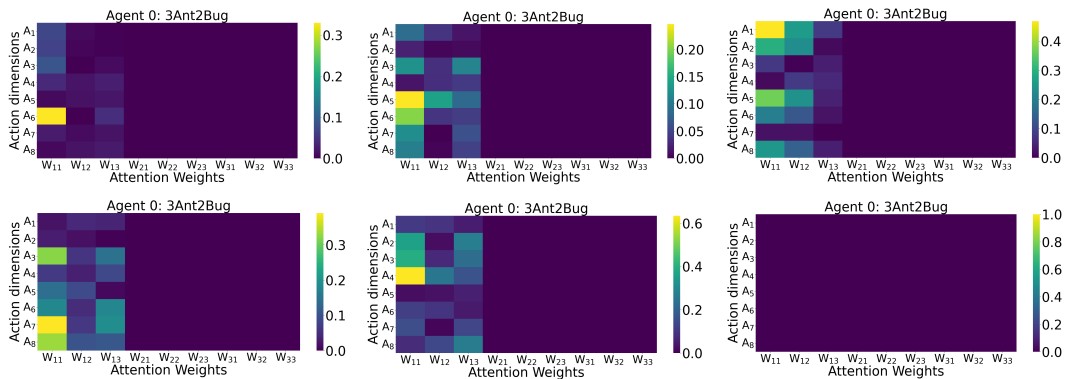

Figure 24: Stage 1 attribution heatmaps for `Ant0` in the `3Ants` vs. `2Bugs` case. The first row corresponds to snapshots 1–3, the second row to snapshots 4–6, tracing how action-level attention weights evolve across timesteps.

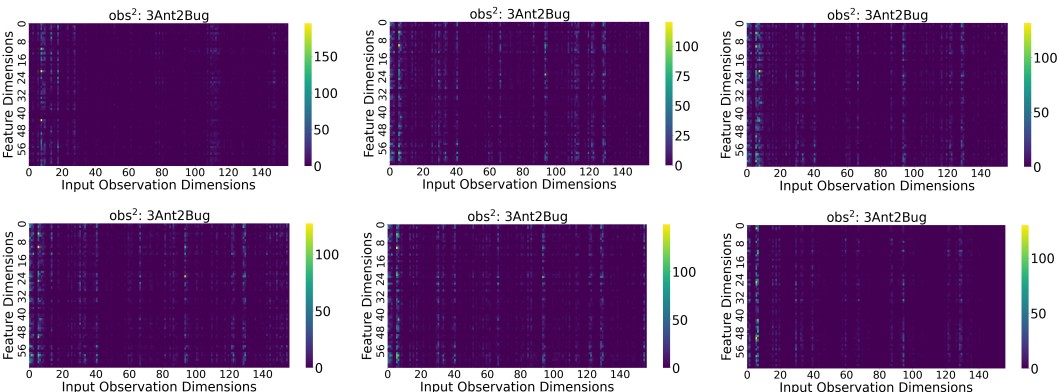

Figure 25: Stage 3 attribution heatmaps for `Ant1`. The first row corresponds to snapshots 1–3, the second row to snapshots 4–6, illustrating how salient attention features are grounded in observation semantics.

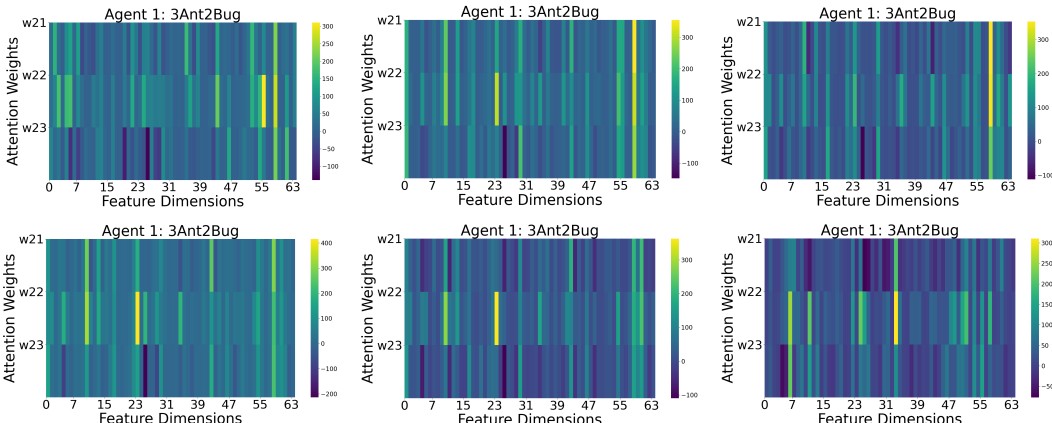

Figure 26: Stage 2 attribution heatmaps for `Ant1`. Snapshots 1–3 are shown in the top row, and snapshots 4–6 in the bottom row, highlighting dominant feature dimensions driving attention allocation.

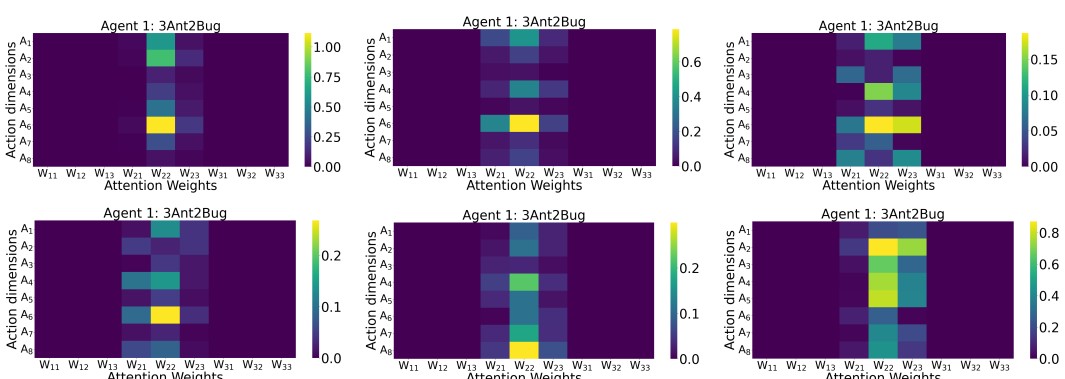

Figure 27: Stage 1 attribution heatmaps for `Ant1`. The first row corresponds to snapshots 1–3, the second row to snapshots 4–6, showing how attention-to-action mappings vary with its central role.

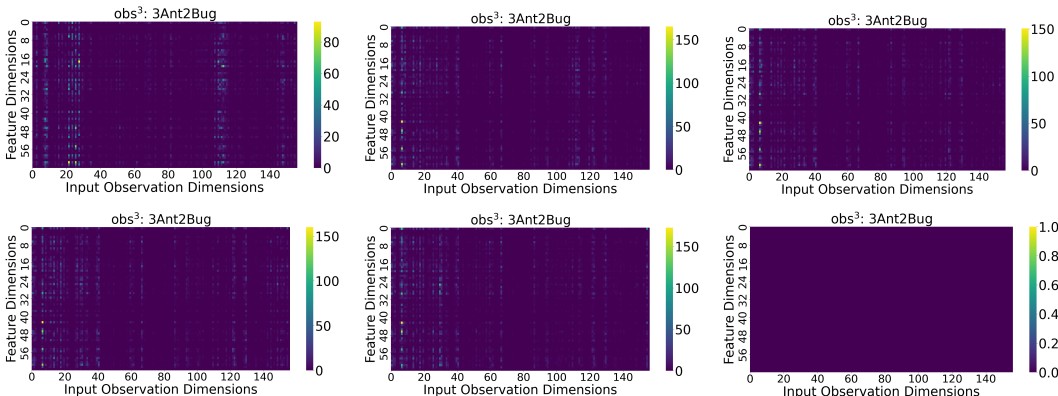

Figure 28: Stage 3 attribution heatmaps for `Ant2`. Snapshots 1–3 (top row) and snapshots 4–6 (bottom row) attribute salient attention dimensions back to observation components.

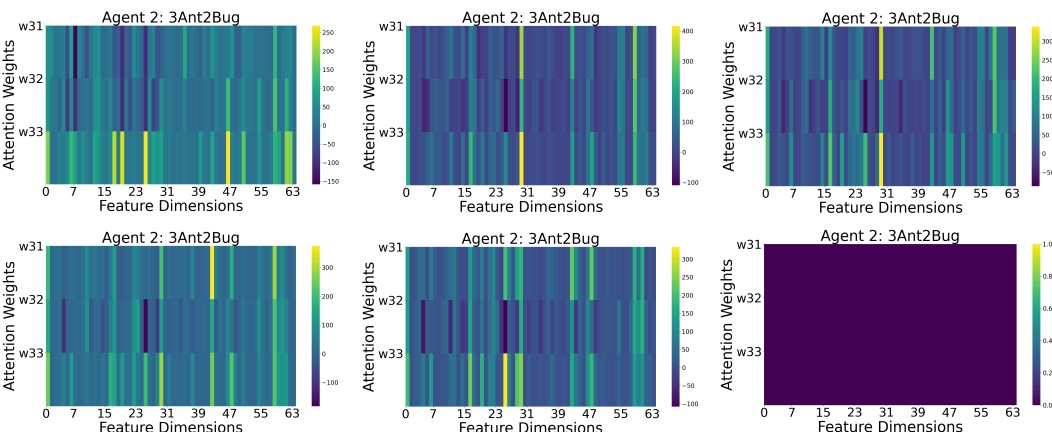

Figure 29: Stage 2 attribution heatmaps for `Ant2`. The first row shows snapshots 1–3 and the second row snapshots 4–6, emphasizing which input features shape its attention over the match.

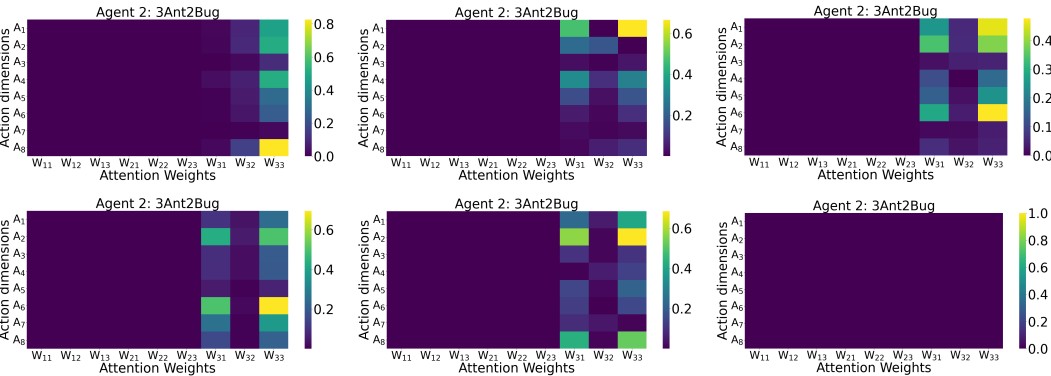

Figure 30: Stage 1 attribution heatmaps for `Ant2`. Snapshots 1–3 (first row) and 4–6 (second row) capture its action-level attention weights over time as the rear-supporting agent.

