# OpenReview forum: "Attention is Advantage: How the Weaker Defeats the Stronger Through Cooperation"
_ICLR.cc/2026/Conference — Submitted to ICLR 2026_

### Official Review · Reviewer_MfZZ · 2025-10-20

**Soundness:** 3
**Presentation:** 3
**Contribution:** 2
**Rating:** 4
**Confidence:** 3

**Summary:**

This paper explores cooperative-competitive mechanisms in multi-agent systems where weaker agents collectively counter stronger opponents. The authors propose a two-phase curriculum learning framework and also leverage transformer-based policy network. They construct a task involving intra-team collaboration and inter-team confrontation, and evaluate the effectiveness of their method. They also conduct some analysis on integrated gradients to attribute the contribution of attention.

**Strengths:**

The paper is well-written. The methods and the experiment details are explained clearly.
The curriculum learning framework is simple yet effective. The investigation on integrated gradients also suggests the explanability of utilizing attention-based architecture of policy network.

**Weaknesses:**

1. One limitation I found is that the proposed curriculum learning method is highly specialized for the bug-ant task proposed in this paper. This makes it unclear whether such a learning strategy is generally applicable to other domains. I expect to see a general algorithm with less human prior knowledge. From this perspective, the current method is relatively restricted.

    Besides, the transformer-based policy architecture seems not novel, which has been proposed and studied for several years.

2. The title seems a bit biased. The authors claim that both curriculum learning and the transformer archiecture are contributions, but the title only highlight the advantage of attention. Besides, I'm not sure whether it is appropriate to use the terms "weaker group", "weaker defeats stronger", etc, since if it the "weaker" can defeat "stronger" in some cases, literally, it is not weaker?

**Questions:**

It is unclear to me why the authors choose to study "weaker groups" overcome "stronger ones", and why it is an important topic. Especially, the proposed method in the paper seems to work for general games involving cooperation within teams and competition between teams.

---

### Official Review · Reviewer_h6CL · 2025-10-23

**Soundness:** 2
**Presentation:** 1
**Contribution:** 2
**Rating:** 2
**Confidence:** 2

**Summary:**

The paper considers multi-agent reinforcement learning problems with two opposing team where one team is larger but with weaker individual members than the other team. The primary focus of learning is to find strategies that can effectively collaborate within the team to beat the other team. The authors focus on a locomotion arena setup with ants and bugs where the goal of the teams is to push the other out of a shrinking circle around a center point.
The authors highlight three main contributions:
1. Transformer-based policy network that can efficiently learn and attribute actions between the team members
2. 2-stage curriculum learning
3. Attribution method to interpret actions
Their empirical results highlight the effects of the first two points through ablation studies while provide explanations of actions using the attribution framework for certain snapshots of games.

**Strengths:**

I found the following points to be compelling of the paper
1. Interesting problem to consider since this is a mixed heterogeneous system with both competitive and cooperative dynamics
2. The authors combine insights from both learning and interpretability
3. They provide an extended discussion of the experimental results to highlight their contributions

**Weaknesses:**

I had a hard time following the paper and might have missed certain points but I found the following to be weaknesses
1. The writing is often imprecise making it hard to follow the paper.
* Notations are not defined such as $\Delta w_{i,j}^{(s,n)}$, $w_{i,j}$, $obs$, orf the difference between $k$ and $\hat{k}$.
* In Section 4.3.3, they introduce "Stage 1" as an Integrated Gradient method without providing citation or background on this method. As someone not familiar with this method, I could not tell if this is a novel method or an adaptation of already existing one.
* It is only described in Section 5 that the goal of the game is to push the opponent out of a shrinking circle. It is only mentioned in previous sections that agents can be eliminated. This would have made understanding the problem and the definition of the reward function in the curriculum learning easier.
* Many discussions (e.g. in Section 4.3.2) are deferred to the Appendix that would be crucial to understand the paper as a whole.
* Most figures are small to read when the paper is printed or have little contrast in black-white.
2. The methods and problem statement seem to be general yet the authors constrain the problem to a single environment that leads questioning the generality of the results.
3. The interpretation method seems to be limited to models with a single attention layer and specific setup as shows on Figure 3. While such network seems to work well on the problem the authors consider, it is not clear whether this scales to larger problem with more agents or larger state/action spaces.
4. The curriculum used for learning is tailor-made for the specific problem limiting the generality of results.

**Questions:**

I have the following questions to the authors to clarify the contribution and results of the paper

Problem Setting (Section 3)
1. What are the observations? The problem statement poses states while in later sections the paper refers to observations. Do agents have full observations about the states and/or actions of all other agents?
2. Are the states and actions continuous or discrete?
3. Do agents in the same team use the same policy or different? The formulation suggests that the share a policy but the problem statement states that they have different action spaces which might not be compatible with a shared policy
4. What is the notion of optimality in this case? Is the desired outcome a Nash Equilibrium or some other optimality?

Learning Method (Section 4)
1. What are the intuitions behind the reward functions defined in the curriculum? The authors state the reward functions but I miss some explanation why they choose them.
2. What embeddings are used in the transformer architecture? It is depicted on Figure 3 but not discussed.
3. What do the following notations denote: $o, \hat{o}, obs, \Delta w$

Experiments (Section 5)
1. The curriculum learning was ablated with MLP policies which later results show are clearly limited. Have the authors ablated curriculum learning with Transformer policies too? That would provide a much stronger evidence for the curriculum method.
2. What is the composite reward function used in the non-curriculum training?
3. Are all policies trained simultaneously regardless of the team?
4. Figure 5 only shows rewards for the Ants. What are the rewards for the Bugs?
5. Could the authors provide some evidence that the training has converged after 1000 epochs? For example on the second figure in Figure 5 shows that the orange line just started shifting when the training finished.
6. How were the PPO hyperparameters tuned and why did the authors decide to use PPO for a multi-agent setting?

---

### Official Review · Reviewer_BqUo · 2025-11-04

**Soundness:** 2
**Presentation:** 2
**Contribution:** 2
**Rating:** 2
**Confidence:** 3

**Summary:**

This paper studies multi-agent reinforcement learning problem with heterogeneous agents that engage in both cooperative and competitive coordination. It proposes a curriculum learning framework that progressively trains agents from basic coordination to complex adversarial scenarios. The approach uses a two-stage training pipeline with a multi-agent transformer model: Stage 1 employs a dense reward to help agents develop stable inward-directed locomotion, while Stage 2 introduces a target-alignment reward that encourages adversarial pursuit and team coordination. To enhance interpretability, the paper performs a post-hoc three-stage attribution analysis that traces causal influence from input observations through attention mechanisms to action outputs. The proposed method outperforms baseline approaches using conventional dense and sparse rewards, and ablation studies show that curriculum-trained agents achieve better coordination and more emergent behaviors. Transformer-based agents also demonstrate consistent superiority over MLP-based architecture.

**Strengths:**

1.	The paper is well written and provides a clear and detailed overview of heterogeneous multi-agent coordination learning in both adversarial settings.
2.	The proposed method introduces a posthoc interpretable pipeline that maps attention to actions, latent vector dimensions to attention weights, and attention scores to observation features, offering valuable insight into which factors drive agent decision-making and emergent behaviors.
3. The visual demonstrations effectively show how Ant agents coordinate for team success over individual survival, realigning dynamically even after one agent is eliminated to continue competing against stronger opponents.
4.	The results show that curriculum learning with a simple reward design can outperform classical methods that rely on more complex, domain-specific reward shaping, highlighting the efficiency of the approach.

**Weaknesses:**

1.	While the idea is conceptually interesting, the paper does not clearly state what specific research question it seeks to investigate, whether it is about improving training stability, understanding attention-driven cooperation, or demonstrating emergent coordination, making the main objective somewhat ambiguous.

2.	Some of the demo videos on the project webpage aren’t accessible. Please ensure that all linked demos are working and clearly associated with the corresponding figures or results.
3.	While the idea of curriculum learning is interesting, the current evaluation is quite limited, focusing only on simple scenarios with a few heterogeneous agents. Although the setup covers three configurations (2A–1B, 3A–1B, 3A–2B), they are all variants of the same arena task with hand-designed rewards (Stage-1: centering; Stage-2: target alignment). The claims would be stronger if the authors (a) tested task reversals (e.g., staying outside while pushing opponents inward), (b) explored different adversarial objectives, and (c) demonstrated whether Stage-1 pretraining transfers to qualitatively different tasks. The current experimental settings leave the generality unclear.

4.	Stage-2 heatmaps highlight different dominant latent feature indices across configurations (e.g., in 3A–2B: features 26, 33, 58 for w₁₁, 11 and 26 for w₂₂, 26 and 58 for w₃₃; in 3A–1B: 29, 44, 60 for w₁₁ and 17 for w₂₂ and w₃₃). What do these differences signify? Are these indices stable up to permutation, or do they represent different functional roles or episode phases? A quantitative correlation or overlap analysis across configurations and episode stages, along with mapping to Stage-3 observation semantics, would make the interpretability much stronger.

5.	All reported populations are small (up to 3 Ants vs. 2 Bugs), even though the text claims the setup “can be generalized to m vs. n.” Evaluating larger or mixed-size teams would test whether cooperative patterns and attention-based attribution signals scale effectively or degrade.

6.	The paper compares curriculum learning (Stage-1 followed by Stage-2) against a non-curriculum composite reward baseline (dense + sparse shaping). A stronger baseline would be to train the non-curriculum baseline using both curriculum rewards (Stage-1 and Stage-2) simultaneously via a weighted sum or an annealing schedule. This would help isolate whether the improvement comes from the curriculum phasing itself or simply from combining multiple reward objectives.

**Questions:**

Please see weaknesses.

---

### Meta-Review · Area_Chair_gcEn · 2026-01-11

**Summary:**

The problem of cooperative-competitive dynamics in heterogeneous multi-agent systems is interesting, but I recommend rejection due to the narrow experimental validation. All experiments are confined to a single arena task with only three small-scale configurations (2v1, 3v1, 3v2), and it remains unclear whether the proposed approach would generalize to larger teams or different task settings. I am also concerned that the interpretability analysis, though conceptually appealing, relies on qualitative narratives without sufficient statistical validation or quantitative rigor. Moreover, the baseline comparisons do not cleanly isolate the curriculum learning contribution, making it difficult to assess the true source of performance gains. Based on the above reasons, I encourage the authors to expand the experimental scope and strengthen the quantitative analysis for future submission.

**Reviewer Concerns:**

The authors are recognized for their effort in developing an interesting multi-agent coordination framework with attention-based interpretability analysis. However, the experimental validation is confined to a single task with small-scale agent configurations, and the interpretability claims lack the quantitative rigor necessary to support the paper's contributions at a venue of this caliber.

Reviewers identified that all experiments occur within one arena task across only three small configurations (2v1, 3v1, 3v2), with no evidence supporting claimed generalization to larger teams or different task objectives. The interpretability analysis provides qualitative narratives but lacks statistical validation, correlation analysis, or explanation for why dominant latent features vary substantially across configurations. Baseline comparisons do not cleanly isolate the curriculum contribution, as curriculum learning is ablated only against MLP policies rather than non-curriculum Transformer policies. Presentation issues including undefined notation, missing citations for the integrated gradients method, and difficult-to-read figures further detract from the submission. These concerns were not addressed during the review period.

**Reviewer Scores:**

The reviewer scores of 2, 2, and 4 indicate strong consensus toward rejection, with two reviewers firmly below the acceptance threshold and the third only marginally so. No reviewer advocated for acceptance, and all expressed fundamental concerns about the narrow experimental scope and limited generalizability of the contributions. The overall sentiment reflects that the submission requires substantially broader validation before it would be suitable for this venue.

---

### Decision · Program_Chairs · 2026-01-26

Reject